# Food Allergy Management Competence in Greek Schools

**DOI:** 10.3390/children10030541

**Published:** 2023-03-11

**Authors:** Gavriela Feketea, John Lakoumentas, Evangelia Papatriantafyllou, Nikolaos Douladiris, Dimitris Efthymiou, Luminita Aurelia Stanciu, Emilia Vassilopoulou

**Affiliations:** 1Department of Pharmacology, Toxicology and Clinical Pharmacology, “luliu Hatieganu” University of Medicine and Pharmacy, 400337 Cluj-Napoca, Romania; fechetea.gabriela@umfcluj.ro; 2Department of Pediatrics, “Karamandaneio” Children’s Hospital of Patra, 26331 Patras, Greece; 3Department of Nutritional Sciences and Dietetics, International Hellenic University, 57400 Thessaloniki, Greece; john.lakoo@gmail.com (J.L.); e.papatriantafyllou@hotmail.com (E.P.); dimitrisefthy@gmail.com (D.E.); vassilopoulouemilia@gmail.com (E.V.); 4Allergy Unit, 2nd Pediatric Clinic, University of Athens, 11527 Athens, Greece; ndouladiris@gmail.com; 5National Heart and Lung Institute, Imperial College London, London W2 1PG, UK

**Keywords:** food allergy, anaphylaxis, school, schoolteachers, adrenaline, adrenaline autoinjector (AAI)

## Abstract

Background: Schoolchildren are likely to consume meals and snacks at school, with a possibility of allergic food reactions and anaphylaxis in the school environment. The school personnel should be informed of the presence of schoolchildren with food allergy (FA) and need to be trained in the management of allergic reactions, as to prepare them to intervene appropriately when necessary. Limited knowledge of FA and its management is documented globally among school staff and there is no uniform protocol in schools. Methods: In this observational cross-sectional study, teachers at state schools throughout Greece completed an online anonymous questionnaire on their awareness of FA reactions and the plans for the management of medical emergencies in their schools of employment. Results: Among the 289 teachers who responded the online invitation, 203 (70.24%) were female and 157 (54%) were aged under 40 years. Females expressed a higher level of concern about the presence of school personnel trained to manage FA symptoms (*p* = 0.001), written instructions, and the availability of adrenaline (epinephrine) at school (*p* < 0.001). A younger age was associated with a higher level of both interest and knowledge on FA management in schools. School directors were more certain about the availability of a special record of children with FA at school (*p* = 0.01), the availability of adrenaline (*p* = 0.006), and written guidelines on the management of serious health incidents at school (*p* = 0.04). Written guidelines instructing children to avoid sharing cutlery, glasses, home-prepared meals, and snacks bought from the school canteen were more common in schools in urban areas (*p* = 0.015). Only 20% of respondents could confirm with certainty that adrenaline autoinjectors (AAIs) were available at their schools, for the purpose of administering to children in the case of a severe FA reaction. Approximately 3/4 of the participating teachers stated that completion of this questionnaire raised their awareness of the risk of FA reactions in children at school. Conclusions: This study, the first in Greece to explore the knowledge of teachers about FA in schoolchildren, revealed the following absences in many schools: a process for identifying children with FA, a written emergency treatment plan, and immediate access to emergency AAI. School FA guidelines are necessary in Greece, and training, which includes the use of AAIs, is required to prepare teachers to manage FA reactions in children at school.

## 1. Introduction

Food allergy (FA) affects up to 4–7% of primary schoolchildren in Europe [1,2]. In Australia, 1 in 20 school-aged children have a proven FA [3] and it affects around 450,000 schoolchildren in Spain [4]. Children spend approximately one third of the day at school and may consume meals and snacks in the school environment, and consequently, 10–18% of allergic/anaphylactic reactions to food may occur at school [4]. Therefore, it is essential to train school staff in the management of allergic reactions [4,5], and to evaluate their knowledge about this health issue both before and after training. This will enable them to intervene appropriately when necessary [6].

Significant limitations have been identified in the competencies of school personnel to manage FA and anaphylactic reactions in the school environment, including the absence of appropriate management plans and failure to recognize and treat reactions with timely administration of adrenaline (epinephrine) [2]. Studies have shown that school teachers largely fail to identify schoolchildren with FA, and have limited knowledge on the causes, symptoms and management of FA and anaphylaxis. In one study conducted by Ercan and colleagues, primary school teachers identified the following as potential causes of allergic reactions: pollen (54%), foods (47%), mites (40%), and medicines (30%). Among the foods they considered responsible for allergy, eggs (30.4%) and strawberries (25.3%) were ranked first and second, while only 7.2% of teachers mentioned tree nuts and peanuts as possible causes of an allergic reaction [7]. In practice, however, the proteins in cow’s milk, eggs, peanuts, tree nuts, wheat, soy, fish, and shellfish account for 90% of all documented FA reactions [8]. Peanut allergy is estimated to affect 1% of all children in the United States (US), and peanut and tree-nut allergies are the leading cause of fatal allergic reactions in children in the US [9,10]. Overall, anaphylaxis in school-age children is caused most commonly by peanuts, tree nuts and cow’s milk, and by food-dependent, exercise-induced anaphylaxis [7]. Rhim and McMorris reported that the most common allergens were milk (81%) and peanuts (62%), followed by nuts (32%), shellfish (28%), egg (23%), wheat (22%), and soy (7%). They also identified other food allergens, including fruit (15%), chocolate (9%), red dye (8%), tomato (8%), fish (6%), orange juice (4%), spices (3%), and cheese (2%). The fruit category included strawberries, blueberries, grapes, mango, pineapple, and orange [11].

The Pediatric Section of the European Academy of Allergy and Clinical Immunology (EAACI), recognizing the limited knowledge among school staff and the lack of uniform protocols in schools, initiated specific action plans to develop protocols and registries containing the appropriate information for managing emergencies relevant to FA reactions and anaphylaxis in the school environment [1]. The guidelines created by EAACI aim to serve as a uniform guide for all centers, but with the ability to be adapted to the specific capabilities of each school [6]. Waserman and colleagues also published guidelines recently, which are intended to be international in scope [12]. However, the implementation of these measures has not been successful, especially in developing countries, as well as in other countries worldwide [1,13,14,15].

In Australia, the Australasian Society of Clinical Immunology and Allergy (ASCIA) recommends that all school staff undertake theoretical anaphylaxis management training every 1–2 years, and maintain regular practice with adrenaline autoinjector (AAI) trainer devices [3,16].

In the US, guideline recommendations have been published by the American Academy of Allergy, Asthma and Immunology (AAAAI) [11]. In the state of Washington, the Office of the Superintendent of Public Instruction Guidelines for Anaphylaxis, published in 2009, states that life-threatening allergy awareness training should be provided to all school personnel each school year. This document proposes that providing FA education to school staff should continue to be an area of emphasis and that school nurses should be given the resources and support they need to achieve this [10].

In a study in Cyprus, an emergency protocol, defined by the Ministry of Health of Cyprus, was considered vital for the provision of first aid by teachers, including the management of severe allergic reactions in the school environment [17].

The purpose of this study was to investigate the preparedness of school teachers in Greece regarding FA and their management in the school environment, based on their self-reported knowledge. Schools in many other countries have been investigated in this way, including Australia, US, Italy, Spain, Turkey, Japan, and Cyprus.

## 2. Materials and Methods

### 2.1. Study Participants

An observational cross-sectional study was conducted between 1st October and 31st December 2022. A total of 8956 school teachers, working in Greek state schools, and participating in school social media networks, were invited to complete an online anonymous questionnaire.

The participants worked as primary or secondary school teachers, special education personnel (SEP), or special support personnel (SSP) in state schools belonging to the Greek educational administration throughout Greece during the academic year 2022–2023. To be included in the analysis, the questionnaires had to be completed up to the final question. Questionnaires not fully completed were excluded.

After the first announcement of the questionnaire in the school social networks, weekly reminders were sent to the members of the school social media networks inviting them to participate in the study.

A detailed informative introductory text preceded the online questionnaire regarding the scope of the study, and an online informed consent was completed before accessing the questionnaire. The study was approved by the Education Department of the 6th Health Region (study ID 3111/18 March 2022) and was conducted in accordance with the code of Ethics of the World Medical Association (Declaration of Helsinki).

### 2.2. Questionnaire

The semi-structured questionnaire used in the EU-funded multidisciplinary Integrated Project EuroPrevall, translated and adapted for use in Greek populations, was used in the study [17,18]. The questionnaire, presented in Table 1, consists of 42 questions, focusing mainly on the awareness of the respondent of FA reactions and the plans for the management of medical emergencies in the school where the respondent was employed. Both open- and closed-ended questions were included, and the questionnaire addressed all members of the educational community. Four demographic variables (gender, age, specialty of the respondent, place of residence) were also recorded. Anonymity and protection of personal data were ensured by the research team.

### 2.3. Statistical Analysis

Data handling and statistical analysis (descriptive and inferential) were performed in R and RStudio. The four categorical demographic variables (gender, age group, teaching specialty, place of residence) were assessed for association with a set of questionnaire response variables that were also categorical. For this reason, descriptive statistics are given in the form of the absolute count and the percent (relative frequency) of any level of each (categorical) variable. The two-tailed Pearson’s chi-squared test of independence was applied to assess each association. Statistical significance was set at *p* ≤ 0.05.

### 2.4. Results

The estimated response rate of the eligible school personnel was 3.2% (*n* = 289), of which 70.24% were females. According to age, 157 (54%) were aged under 40 years, and only 47 (16.26%) were aged over 50 years and 5 (1.73%) over 60 years.

The majority of the participants were primary school teachers (39.79%), followed by secondary school teachers (28.72%). School principals comprised 5.88%, SEP 18.34%, and SSP 7.27% of the study sample. The higher response rates were recorded in Western Macedonia (*n* = 125, 43.25%) and Central Macedonia (*n* = 58, 20.07%), and the rest of the respondents were distributed throughout the other regions of Greece. Table 2 shows the demographic characteristics of the study participants. Approximately 30.4% (*n* = 88) lived in an urban area, whereas the rest lived in a semi-urban or rural area.

Of the 289 participants, 101 (34.95%) reported that the school where they were employed had in place a process for identifying students with chronic health conditions or special medical needs, and 89 (30.8%) reported no such process. A large number of the participating teachers (44.64%) reported not being informed if children with FA are identified on a separate list at school. Only 18.69% of the teachers were aware of the existence of protocols, circulars, or written guidelines on how serious health incidents should be dealt with at school. Only 15.22% believed that the school staff in their schools is trained to recognize the signs and symptoms of FA, whereas more than a half considered that the staff is not trained. However, 61.59% reported that they know the symptoms that could be associated with FA. Almost half (48.79%) of the respondents admitted to being not informed on food labels and possibly hazardous ingredients.

The study participants exhibited several gender-dependent differences in FA awareness, preparedness, and attitudes, as shown in Table 3. Specifically, more women than men were aware of the presence of a child with allergy in the school during the last 3 years (*p* = 0.05), and were informed on the process for identifying children with chronic conditions or a special medical need (*p* < 0.001). Additionally, women were more often informed about a record kept in the school of children with FA (*p* = 0.03), and knew the possible symptoms in the event of a FA reaction (*p* < 0.001). Females more often knew of the presence of the school personnel members trained to manage FA symptoms (*p* = 0.001), and about the availability of adrenaline autoinjectors (referred to in the questionnaire as Anapen) at school (*p* < 0.001) and were aware of written instructions for children to avoid sharing cutlery, glasses, home-prepared meals, or snacks bought from the school canteen (*p* = 0.001). Overall, females indicated a higher level of concern about the incidence of FA at school (*p* < 0.001); they more often considered that it is a serious health issue (*p* = 0.01) and stated that their participation in the study motivated them to become more informed on the risks and management of FA (*p* = 0.04).

The age of the participants appeared to influence their interest and knowledge about FA management at schools. Specifically, younger participants (<40 years) were less aware of the process for identifying schoolchildren with chronic disease or special medical needs (*p* = 0.006). Conversely, they were more often informed about special records for children with FA at school (*p* = 0.03), could recognize FA symptoms at a higher rate (67.52%) (*p* = 0.03), and were aware of the availability of adrenaline (*p* = 0.03) and written guidelines on the management of serious health incidents at school (*p* < 0.001) (Table 4).

Unexpectedly, little difference was identified in FA preparedness and knowledge between school directors and the other school staff. Specifically, school directors were more certain about the availability or not of a special record for children with FA at school (*p* = 0.01) the availability or not of adrenaline (*p* = 0.006) and the existence of written guidelines on the management of serious health incidents at school (*p* = 0.04).

No significant differences were detected on knowledge and preparedness between participants who lived in urban or semi-urban and rural areas, apart from the awareness of written guidance for children to avoid sharing cutlery, glasses, home-prepared meals, or snacks bought from the school canteen, which was higher in urban areas (*p* = 0.015).

Almost half of the respondents did not know whether adrenaline autoinjectors (AAIs) were available at their school to be administered for severe FA reactions, and only 20% could confirm their presence with certainty. Regarding knowledge of any FA awareness organization, 93.43%% of the respondents were unable to identify a such an organization. The responses of teachers on the questionnaire are shown in detail in Table 3.

Finally, approximately 3/4 of the participants stated that they believe that this interview will motivate them to reappraise the risk of food reactions in children at school.

## 3. Discussion

The results of this study confirm findings from previous investigations conducted in other countries, which demonstrated that women were more willing to respond to questionnaires about teachers' preparedness to handle a child with allergies at school [2,4]. It also appears from their responses that women are more interested in participating actively in the management of FA reactions in school. The study questionnaire was distributed through social media networks and completed online, which could be an explanation for the response rate being inversely dependent on age, as younger people are, in general, more comfortable with social media participation. As in this study, previous studies have identified a lack in FA education, symptom recognition and management plans in schools [19]. However, to the best knowledge of the authors, there have been no relevant published data from Greece to date.

As this study was cross-sectional, its design was restricted to observing and recording the current situation, without the possibility of evaluating the potential of intervention in real-life school settings. Multivariate analysis was not applied, as causality and prediction among the participants were not an aim of the study.

### 3.1. Knowledge of Teachers about the Management of FA

Despite the recognition of its potential dangers, adrenaline is the standard treatment for acute anaphylaxis [20], at a dosage based on the child’s weight, and usually administered in the form of an AAI [21,22]. Antihistamines, sublingual isoproterenol, inhaled epinephrine, and corticosteroids fail to prevent or relieve severe anaphylactic reactions [23,24], and their use for first-line treatment or prevention of anaphylaxis is considered inappropriate [24,25]. Antihistamines may help relieve skin symptoms and are usually given only to prevent a later phase reaction, but research findings do not currently support their use in the acute stage [24,26].

In a similar Turkish study, 93.7% of participants reported not knowing the initial drug to be used in the event of a severe allergic reaction, whereas those who thought they knew reported antihistamines and/or salbutamol [7]. The present study suggests that teachers are not knowledgeable about the presence in their own schools of guidelines for the prevention and management of anaphylaxis or about the availability of AAIs. None of the teachers mentioned adrenaline as the first drug to be used for anaphylaxis, only 10% knew about self-injection of adrenaline/epinephrine, and only 4% knew how and where to administer it [7]. It appears that Scottish schools are unprepared to deal with severe allergic reactions, as only 12% have access to AAIs and less than half have documented evidence of a trained teacher on staff [27]. Other European countries show a similar deficient pattern of school preparedness for FA reactions [28].

The present study showed that Greek school teachers are not well informed about anaphylaxis. In other studies, the majority of teachers were unable to recognize anaphylaxis symptoms, lacked knowledge of AAIs, lacked confidence in providing emergency first aid during anaphylactic reactions, and were unaware of their school's anaphylaxis management action plan [7,17]. In most cases, the responsibility for managing schoolchildren with FA rests with the school staff, and lack of knowledge and recognition of anaphylaxis is a major concern for school staff, who consider that information should be provided at school level [11]. Deficiencies are documented in the knowledge and management of allergic reactions by the school staff in daily contact with children with FA who are at risk of a reaction. The main limitations noted are in the recognition of the symptoms of an allergic reaction, and delay in the administration of treatment, specifically adrenaline [29,30]. Studies designed to assess the self-efficacy of school personnel have revealed serious deficiencies in the management of FA and the treatment of allergic reactions in schoolchildren and indicate a need for special educational intervention to ensure the appropriate management of allergic reactions in children with FA [2,5,6,7]. A Japanese study of school nurses, teachers, and care workers involved with children who had been prescribed AAI showed that hands-on training resulted in a dramatic improvement in self-efficacy [31]. Italian studies reported the effectiveness of specific multidisciplinary training courses in improving the self-efficacy of school staff in managing FA and anaphylaxis [2,5,32].

The problem is global in nature and findings from studies using a variety of methods show that the situation in schools is far from safe for children with asthma or FA [6]. A nurse educator providing hands-on AAI use demonstrations and training was shown to be an effective strategy in helping teachers confidently recognize and treat severe allergic reactions. These positive effects were independent of prior training and sustained for up to 12 months after the presentation [10].

### 3.2. Action Plans, Guidelines and Laws

Various studies indicate that the majority of teachers do not know if there are action plans and protocols for FA at their schools [6,7,11,33]. Almost all the teachers in one study (96.8%) expressed interest in an intervention protocol on how to act in the case of a child with an asthma attack or an anaphylactic reaction at school [6]. All children with FA should have an individualized health plan that includes a written emergency plan, which is shared with their schools [24].

Several scientific and governmental organizations promote the development of (a) school policies (led by the school principal/headteacher) for managing children with allergies in school, in collaboration with a multi-professional team of experts, and (b) establishment of a strong communication network between all stakeholders [21,30,34,35]. In the UK, regulations are region specific; England, Wales, Scotland, and Northern Ireland each have guidance documents produced by different branches of government [3,16].

In 2005, the Ontario government, following a fatal anaphylactic reaction at school, passed the “Anaphylactic Student Protection Act” (Sabrina’s Law), which was the first legislation of its type in the world, requiring school principals to collaborate with health professionals to develop and maintain an anaphylaxis policy in each school [36]. This legislation has improved the efforts made by schools to support schoolchildren with FA, although it is apparent that further resources may be needed to prepare school staff to manage anaphylaxis [28,37].

All Australian states and territories have anaphylaxis policies and guidelines that outline the management of anaphylaxis in schools, and Australian school systems are legally required to equip staff with the skills to correctly identify and manage anaphylaxis [3]. In Italy, where there are no nurses on the school staff, the management of FA in schoolchildren is undertaken by the school staff. Polloni and colleagues found that 79.3% of school staff were able to identify the foods most likely to cause FA, 90.8% knew most of the common symptoms of FA, and 81.9% were familiar with typical symptoms of anaphylaxis. These findings are encouraging, but this is one of the few studies that demonstrates a high level of knowledge and familiarity with FA among school personnel [5]. 

Although there are no nurses in most state schools in Greece, in an effort to improve school care for children with health problems, nurses may be placed in the school environment, depending on the local capacity and the recommendation of an allergist or pediatric allergist, but usually the teachers have to take care of the children in need, which poses a significant demand. The guidelines for teachers, issued by Greek Institute of Child Health, state that in the case of children suffering from serious anaphylactic reactions, a written declaration of consent is required from the parents/guardians for the provision of first aid to the children at school. The appropriate medication and written instructions from the allergist or pediatric allergist for its administration are provided by the parents for each child [38]. The UK, US, and Australia have passed legislation to allow supply of AAIs to schools [3,39]. The UK law now allows school staff to administer emergency AAI to any child with symptoms of anaphylaxis [28].

### 3.3. Intervention Programs

Internationally, there is no example of mandatory country-level anaphylaxis training for school staff, although many countries have introduced voluntary guidelines [3]. In Greece, educational seminars on managing child with allergies at school are organized sporadically, but training is not mandatory. Similarly, in Italy, the teachers attend free courses on FA and anaphylaxis management at school on a voluntary basis. These courses include knowledge and thoughts and feelings about FA [2].

Juliá-Benito and colleagues describe the pioneering programs in Spain that have been developed by the educational services and emergency health care services in the Autonomous Communities of Andalusia and Galicia. The School Alert program, with the main objective of ensuring prompt and effective care for schoolchildren experiencing life-threatening emergencies due to anaphylaxis, seizures, or diabetes mellitus-related hypoglycemia, was launched in March 2007 in Galicia, and introduced in the Balearic Islands in 2014. In this program, parents enter a child with a life-threatening chronic illness on the 061 urgent care service register. In the event of an emergency, the school notifies 061, where a doctor provides instructions pending the arrival of medical personnel. In addition, school teachers receive training in the management of the life-threatening illnesses included in the program [6].

In Japan, workshops have been held for school staff on the management of children with life-threatening allergies, consisting of three sessions: a presentation of anaphylaxis, hands-on training on AAI administration using training devices, and a question-and-answer session. The lecture topics included the mechanisms, signs and symptoms of FA, prevention measures against accidental exposure to allergens, and medical management of adverse reactions [31]. Many of the school staff involved in the workshops had limited prior experience with anaphylaxis and demonstrated poor self-efficacy prior to the workshop. The finding is consistent with those of studies in Turkey and California [7,31,40]. In the US, a training program is conducted by a licensed registered nurse who is a Food Allergy Educator. The program teaches school staff to recognize the signs and symptoms of allergic reactions, how to use AAIs, and provides other basic information about food allergies [10].

According to the ASCIA guidelines, Victoria and New South Wales require theory training every 2 years, whereas other jurisdictions in Australia require training less frequently or at unspecified intervals [16]. Although some jurisdictions require all school staff to undergo theoretical training, others require only designated staff (e.g., subject teachers who supervise at-risk children) to undergo training [3,16]. All Australian jurisdictions require children at risk of anaphylaxis to have an individual written health care/management plan, including an individual emergency response plan, which is completed and signed by a medical practitioner [3,16].

In Greece, school principals, and teachers in general, are encouraged to request relevant advice from allergists and pediatricians, or other allied health professionals (AHP) with experience in dealing with children with allergies. Guidance from experienced dietitians or nurses and support from psychologists and other AHPs are essential for the effective management of FA in schools [41]. Under ideal circumstances, teachers should receive instruction from a multidisciplinary team on allergic reactions in children, and specific training on how to recognize the signs and symptoms of FA and apply the appropriate management. The use of a non-validated questionnaire may limit the assessment of whether schools have action plans for the management of FA and anaphylaxis [7], but relevant knowledge can be assessed in a variety of ways, including through questionnaires.

The present study provides information that reveals the need for the implementation in Greece of an information and training policy for teachers in order to improve the safety of children with FA in the school environment. The low response rate is a significant limitation of the study and indicative of the low engagement of the teachers with the management of FA. The study questionnaire was distributed and completed online, which could also explain the low response rate, as potential respondents had no previous information on the study to increase their interest and motivation to respond.

## 4. Conclusions

This study is the first in Greece to explore the knowledge of school teachers on FA and its management. It revealed a variety of gaps and difficulties among teachers in identifying and managing FA reactions and anaphylaxis in children attending their schools.

The absence of a specific process for identifying schoolchildren with FA, the lack of a written emergency treatment plan and of immediate access to emergency AAI are only few of the important deficiencies reported.

The findings of this study highlight the absence of structured school-wide staff training on FA in Greece. The teachers responding to this study considered that even this basic completion of a questionnaire provided motivation for them to reconsider the risk of FA reactions in the children at their schools.

This study confirms the importance of implementing organized training programs for teachers, with the participation of stakeholders from the healthcare and education systems, in order to prepare teachers to effectively manage FA in schoolchildren and to ensure a safe environment in schools. The scarcity of organized frameworks for management of FA in schools is a global problem, as demonstrated by the publication of studies from widely different geographical regions. Our study is the first to describe the current situation in Greek schools, and its findings emphasize the importance of implementing systematic intervention programs aimed at informing, sensitizing, and educating teachers about food allergies. This includes recognizing allergy symptoms and reacting appropriately in the event of an anaphylactic reaction in children at school, with the official support of relevant public health and education sectors.

## Figures and Tables

**Table 1 children-10-00541-t001:** Questionnaire on food allergy for school personnel.

What is your position in the school? How many people work at this school? Do you know if any schoolchildren have presented a food allergic reaction during the past three years? If the previous answer is YES, how many children? Do you know in what food allergic reactions have occurred? Is there a school protocol regarding children with anamnesis of chronic medical conditions, like allergies? How are food-allergic children identified? Do you keep records of them? If the previous answer is YES, who is authorized to access in such information? Do you update the records? What type of data do the records include? Are you familiar with the symptoms of food allergy? If the previous answer is YES, can you list them? Are teachers and the rest of the personnel informed and trained to recognize food allergens and symptoms? Has an educational program been offered to them? What was the nature of the offered educational program and in how many sessions? What is the school protocol for the treatment of a severe anaphylactic reaction? What would your first action be? (A) contact the parents (B) remain to see how it goes (C) contact child’s family doctor (D) call emergency (E) administrate Anapen (since Anapen is the only available EIA in Greece, the brand name was used)? If you choose more than one of the above actions, can you place them in a row from the first to the last? Do you have Anapen available at school for severe food allergy reactions? If the previous answer is YES, where do you store it? What is your estimation on the percentage of teachers and non-teaching personnel that are familiar with Anapen’s availability? Does the school personnel know how to use Anapen? Is there a specially trained person to administrate Anapen in the case of anaphylaxis? Has the Greek Ministry of Education, Culture, Sport and Youth provided Directions on how to treat anaphylaxis or other food-related clinical features? Do you have any instructions on children to avoid sharing cutlery, glasses, home-prepared meals or snacks they buy from the school canteen? If YES, what is your estimation on the percentage of school personnel that know these instructions? Are children trained to follow such instructions? Do the parents know about that school instructions? Do you believe that school rules, forbiting to share things/foods, can negatively affect children’s personality and relationship? Is the school personnel informed on food labels and questionable ingredients? If the previous answer is YES, in what percentage? Do children get informed about food labelling as a part of their school education? Do you consider that you focus a lot on that health issue and thus resulting in a group of foods get excluded? Do you know any food allergy awareness organization? What is its name? Does the school get in contact with that organization? If the previous answer is YES, how much useful is? On a scale of 0 to 100, how much do you worry about the incidence with food allergies at school? If it near 100/or lower of 50, why? Do you give more or less sententiousness compared with other health problems? Do you think that this interview will motivate you to reconsider the risk of food allergy reactions? What type of actions would you propose specialists to offer at your school in order to raise awareness on food allergy?

**Table 2 children-10-00541-t002:** Characteristics of study population of Greek school staff (N = 289).

GENDER	
Male	86 (29.76%)
Female	203 (70.24%)
AGE (years)	
22–29	79 (27.34%)
30–39	78 (26.99%)
40–49	80 (27.68%)
50–59	47 (16.26%)
60–67	5 (1.73%)
OCCUPATION	
Director	17 (5.88%)
Special Support Staff (SSP)	21 (7.27%)
Special Education Personnel (SEP)	53 (18.34%)
Secondary Education Teacher (SET)	83 (28.72%)
Primary Education Teacher (PET)	115 (39.79%)
RESIDENCE	
Urban	88 (30.4%)
Semi-urban, Rural	201 (69.6%)
REGION OF GREECE	
Eastern Macedonia and Thrace	8 (2.77%)
Attica	28 (9.69%)
North Aegean	1 (0.35%)
Central MacedoniaWest Greece	2 (0.69%)
West Macedonia	125 (43.25%)
Epirus	8 (2.77%)
Thessaly	10 (3.46%)
Ionian Islands	9 (3.11%)
Central Macedonia	58 (20.07%)
Crete	9 (3.11%)
Southern Aegean Sea	23 (7.96%)
Peloponnese	5 (1.73%)
Central Greece	3 (1.04%)

**Table 3 children-10-00541-t003:** Gender differences in responses of school staff to questionnaire on food allergy (FA) knowledge and policy.

		Male = 86 (29.76%)	Female = 203 (70.24%)	*p* Value
Q1 = Allergy event				*0.05*
Yes	138 (47.75%)	33 (38.37%)	105 (51.72%)	
No	151 (52.25%)	53 (61.63%)	98 (48.28%)	
Q2 = Identifying children with chronic disease				*<0.001*
I do not know	99 (34.26%)	44 (51.16%)	55 (27.09%)	
Yes	101 (34.95%)	18 (20.93%)	83 (40.89%)	
No	89 (30.8%)	24 (27.91%)	65 (32.02%)	
Q3 = Record of FA children				*0.03*
I do not know	129 (44.64%)	48 (55.81%)	81 (39.9%)	
Yes	95 (32.87%)	25 (29.07%)	70 (34.48%)	
No	65 (22.49%)	13 (15.12%)	52 (25.62%)	
Q4 = FA symptoms				*<0.001*
Yes	178 (61.59%)	32 (37.21%)	146 (71.92%)	
No	111 (38.41%)	54 (62.79%)	57 (28.08%)	
Q5 = School personnel training on FA symptoms				*0.001*
I do not know	92 (31.83%)	41 (47.67%)	51 (25.12%)	
Yes	44 (15.22%)	9 (10.47%)	35 (17.24%)	
No	153 (52.94%)	36 (41.86%)	117 (57.64%)	
Q7 = ANAPEN available				*<0.001*
I do not know	137 (47.4%)	56 (65.12%)	81 (39.9%)	
Yes	66 (22.84%)	16 (18.6%)	50 (24.63%)	
No	86 (29.76%)	14 (16.28%)	72 (35.47%)	
Q8 = Written guidelines				0.591
I do not know	169 (58.48%)	53 (61.63%)	116 (57.14%)	
Yes	54 (18.69%)	13 (15.12%)	41 (20.2%)	
No	66 (22.84%)	20 (23.26%)	46 (22.66%)	
Q9 = Borrowing food/ cutlery				*0.001*
I do not know	113 (39.1%)	46 (53.49%)	67 (33%)	
Yes	96 (33.22%)	17 (19.77%)	79 (38.92%)	
No	80 (27.68%)	23 (26.74%)	57 (28.08%)	
Q10 = School personnel training on food labels				0.073
I do not know	112 (38.75%)	42 (48.84%)	70 (34.48%)	
Yes	36 (12.46%)	9 (10.47%)	27 (13.3%)	
No	141 (48.79%)	35 (40.7%)	106 (52.22%)	
Q11 = Organization on FA awareness				0.936
Yes	19 (6.57%)	5 (5.81%)	14 (6.9%)	
No	270 (93.43%)	81 (94.19%)	189 (93.1%)	
Q12 = Level of concern scale				*<0.001*
0–25%	88 (30.45%)	46 (53.49%)	42 (20.69%)	
25–50%	76 (26.3%)	18 (20.93%)	58 (28.57%)	
50–75%	75 (25.95%)	16 (18.6%)	59 (29.06%)	
75–100%	50 (17.3%)	6 (6.98%)	44 (21.67%)	
Q13 = FA important health concern				*0.01*
More	169 (58.48%)	40 (46.51%)	129 (63.55%)	
Less	120 (41.52%)	46 (53.49%)	74 (36.45%)	
Q14 = Interest on FA				*0.04*
Yes	219 (75.78%)	58 (67.44%)	161 (79.31%)	
No	70 (24.22%)	28 (32.56%)	42 (20.69%)	

**Table 4 children-10-00541-t004:** Age differences in responses of school staff to questionnaire on food allergy (FA) knowledge and policy.

	Age < 40 [N = 157]	Age > /40 [N = 132]	*p* Value
Q1 = Allergy event			0.404
Yes	79 (50.32%)	59 (44.7%)	
No	78 (49.68%)	73 (55.3%)	
Q2 = Identifying children with chronic disease			*0.006*
I do not know	**62 (39.49%)**	**37 (28.03%)**	
Yes	59 (37.58%)	42 (31.82%)	
No	36 (22.93%)	53 (40.15%)	
Q3 = Record of FA children			*0.003*
I do not know	72 (45.86%)	57 (43.18%)	
Yes	**61 (38.85%)**	**34 (25.76%)**	
No	24 (15.29%)	41 (31.06%)	
Q4 = FA symptoms			*0.03*
Yes	**106 (67.52%)**	**72 (54.55%)**	
No	51 (32.48%)	60 (45.45%)	
Q5 = School personnel training on FA symptoms			0.23
I do not know	49 (31.21%)	43 (32.58%)	
Yes	29 (18.47%)	15 (11.36%)	
No	79 (50.32%)	74 (56.06%)	
Q7 = ANAPEN available			*0.03*
I do not know	79 (50.32%)	58 (43.94%)	
Yes	**41 (26.11%)**	**25 (18.94%)**	
No	37 (23.57%)	49 (37.12%)	
Q8 = Written guidelines			*<0.001*
I do not know	93 (59.24%)	76 (57.58%)	
Yes	**40 (25.48%)**	**14 (10.61%)**	
No	24 (15.29%)	42 (31.82%)	
Q9 = Borrowing food/ cutlery			0.225
I do not know	58 (36.94%)	55 (41.67%)	
Yes	59 (37.58%)	37 (28.03%)	
No	40 (25.48%)	40 (30.3%)	
Q10 = School personnel training on food labels			0.165
I do not know	65 (41.4%)	47 (35.61%)	
Yes	23 (14.65%)	13 (9.85%)	
No	69 (43.95%)	72 (54.55%)	
Q11 = Organization on FA awareness			0.299
Yes	13 (8.28%)	6 (4.55%)	
No	144 (91.72%)	126 (95.45%)	
Q12 = Level of concern scale			0.307
0–25%	41 (26.11%)	47 (35.61%)	
25–50%	42 (26.75%)	34 (25.76%)	
50–75%	43 (27.39%)	32 (24.24%)	
75–100%	31 (19.75%)	19 (14.39%)	
Q13 = FA important health concern			1
More	92 (58.6%)	77 (58.33%)	
Less	65 (41.4%)	55 (41.67%)	
Q14 = Interest on FA			0.674
Yes	121 (77.07%)	98 (74.24%)	
No	36 (22.93%)	34 (25.76%)	

## Data Availability

No new data were created.

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
