# Peer review of "Food Allergy Management Competence in Greek Schools"

_children, 2023, doi:10.3390/children10030541_

Round 1

Reviewer 1 Report

This is an observational cross-sectional study conducted over a period of 3 months. The aim of this study was to investigate the preparedness of the school environment in Greece regarding food allergies. In my opinion, it would have been most useful to teach them first and then to investigate their knowledge.
Lines 96-99... I recommend you to move them to materials and methods.
Line 105... please reformulate "The inclusion criteria included"
You did not mention the exclusion criteria.
You did not mention the limits of the study.
You have not mentioned anything about informed consent, neither any Ethical approval.
Line 498...the last reference is not complete.
The references are appropriate, the article presents 41 references, being current (6 references from the last 2 years)

Author Response

Thank you for your overall comment. We conducted the study in order to screen their current knowledge and afterwards to design proper intervention to increase their knowledge and skills on how to manage the child with allergy at school.

Lines 96-99... I recommend you to move them to materials and methods.

Thank you for your comment. We have decided to remove this sentence, as it does not add to the content of our research.

Line 105... please reformulate "The inclusion criteria included"

You did not mention the exclusion criteria.

You did not mention the limits of the study.

You have not mentioned anything about informed consent, neither any Ethical approval.

All the above information is added (lines 103-115 and 371-376). Thank you.

Line 498...the last reference is not complete.

Thank you. We completed this reference.

The references are appropriate, the article presents 41 references, being current (6 references from the last 2 years)

Thank you for your comment.

Reviewer 2 Report

The authors conducted an cross-sectional study to assess the knowledge of Greek teachers regarding management of food allergy in children at their schools. The manuscript is well-written. I only have one minor comment. Can the authors please elaborate on the implications their results further in the discussion section and perhaps propose some specific interventions to improve school staff's preparedness for managing food allergies that occur in their students.

Author Response

We appreciate your overall comment. A paragraph on future intervention is added at the end of the manuscript (lines 371-376).

Reviewer 3 Report

Thank you very much for sharing this article with me. This is a well written piece. I have several minor comments.

My main concern is that the authors did not employ multivariate statistics. We do not know if shown results are net of other factors. This has several implications. For example, although the authors provide some context of interventional programs, I am not sure if results in this study can be used to point to some intervention recommendations, especially they are only showing gross impacts. Thus, it may be important to employ multivariate analysis if sample size allows. If not, this can be described as part of limitation section.

Author Response

Thank you for your overall comment.

We appreciate your suggestion. After thorough discussion with the statistician, we have decided that a multivariate analysis will not the current study, due to the following reasons: We employed a hypothesis testing procedure for several bivariate associations of variables, but not a multivariate approach, mostly due to the following reasons. We actually grouped the available variables into two groups (A and B), and aimed to associate all of the variable couples from different groups. The (categorical) variables of group A were all of demographic nature (gender, age group, location, exact role in the school), while the variables of group B were questionnaire variables, either scale or categorical. Due to the nature of the variables of group A and in our hypothesis testing procedure, neither we meant that "association implies causation", nor the process aimed to predict any variable. As a consequence and to our opinion, a multivariate approach such as multiple linear or logistic regression would be meaningless if we used group A variables as dependent, which would be the more logical approach. On the other hand, if we used group B variables as dependent, a model would have to run for each separate questionnaire variable, either a linear or a logistic or a multinomial logistic model, a fact that would produce a vast amount of outcomes that would confuse the reader and would not add significantly to the outcomes.

Round 2

Reviewer 1 Report

The authors improved the structure of the article. It would have been easier to identify the additions if they have been written in a different color. I hope that the authors keep their promise and design proper intervention to increase tneir knowledge and skills on how to manage the child with allergy at school.